# ON ORTHOGONALITY AND LEARNING RECURRENT NETWORKS WITH LONG TERM DEPENDENCIES

**Eugene Vorontsov [1,2], Chiheb Trabelsi [1,2], Samuel Kadoury [1,3], Chris Pal [1,2]**
[1] École Polytechnique de Montréal, Montréal, Canada
[2] Montreal Institute for Learning Algorithms, Montréal, Canada
[3] CHUM Research Center, Montréal, Canada
{eugene.vorontsov, chiheb.trabelsi,
samuel.kadoury, christopher.pal}@polymtl.ca

## ABSTRACT

It is well known that it is challenging to train deep neural networks and recurrent neural networks for tasks that exhibit long term dependencies. The vanishing or exploding gradient problem is a well known issue associated with these challenges. One approach to addressing vanishing and exploding gradients is to use either soft or hard constraints on weight matrices so as to encourage or enforce orthogonality. Orthogonal matrices preserve gradient norm during backpropagation and can therefore be a desirable property; however, we find that hard constraints on orthogonality can negatively affect the speed of convergence and model performance. This paper explores the issues of optimization convergence, speed and gradient stability using a variety of different methods for encouraging or enforcing orthogonality. In particular we propose a weight matrix factorization and parameterization strategy through which we can bound matrix norms and therein control the degree of expansivity induced during backpropagation.

## 1 INTRODUCTION

The depth of deep neural networks confers representational power, but also makes model optimization more challenging. Training deep networks with gradient descent based methods is known to be difficult as a consequence of the vanishing and exploding gradient problem (Hochreiter & Schmidhuber, 1997). Typically, exploding gradients are avoided by clipping large gradients (Pascanu et al., 2013) or introducing an $L_2$ or $L_1$ weight norm penalty. The latter has the effect of bounding the spectral radius of the linear transformations, thus limiting the maximal gain across the transformation. Krueger & Memisevic (2015) attempt to stabilize the norm of propagating signals directly by penalizing differences in successive norm pairs in the forward pass and Pascanu et al. (2013) propose to penalize successive gradient norm pairs in the backward pass. These regularizers affect the network parameterization with respect to the data instead of penalizing weights directly.

Both expansivity and contractivity of linear transformations can also be limited by more tightly bounding their spectra. By limiting the transformations to be orthogonal, their singular spectra are limited to unitary gain causing the transformations to be norm-preserving. Le et al. (2015) and Henaff et al. (2016) have respectively shown that identity initialization and orthogonal initialization can be beneficial. Arjovsky et al. (2015) have gone beyond initialization, building unitary recurrent neural network (RNN) models with transformations that are unitary by construction which they achieved by composing multiple basic unitary transformations. The resulting transformations, for some n-dimensional input, cover only some subset of possible $n \times n$ unitary matrices but appear to perform well on simple tasks and have the benefit of having low complexity in memory and computation.

The entire set of possible unitary or orthogonal parameterizations forms the Stiefel manifold. At a much higher computational cost, gradient descent optimization directly along this manifold can be done via geodesic steps (Nishimori, 2005; Tagare, 2011). Recent work (Wisdom et al., 2016) has proposed the optimization of unitary matrices along the Stiefel manifold using geodesic gradient descent. To produce a full-capacity parameterization for unitary matrices they use some insights

from Tagare (2011), combining the use of a canonical inner products and Cayley transformations. Their experimental work indicates that full capacity unitary RNN models can solve the copy memory problem whereas both LSTM networks and restricted capacity unitary RNN models having similar complexity appear unable to solve the task for a longer sequence length ($T = 2000$).

In contrast, here we explore the optimization of real valued matrices within a configurable margin about the Stiefel manifold. We suspect that a strong constraint of orthogonality limits the model's representational power, hindering its performance, and may make optimization more difficult. We explore this hypothesis empirically by employing a factorization technique that allows us to limit the degree of deviation from the Stiefel manifold. While we use geodesic gradient descent, we simultaneously update the singular spectra of our matrices along Euclidean steps, allowing optimization to step away from the manifold while still curving about it.

## 1.1 VANISHING AND EXPLODING GRADIENTS

The issue of vanishing and exploding gradients as it pertains to the parameterization of neural networks can be illuminated by looking at the gradient back-propagation chain through a network.

A neural network with $n$ hidden layers has pre-activations

$$\mathbf{a}_i(\mathbf{h}_{i-1}) = \mathbf{W}_i \, \mathbf{h}_{i-1} + \, \mathbf{b}_i, \;\; i \in \{2, \cdots, n\} \tag{1}$$

For notational convenience, we combine parameters $\mathbf{W}_i$ and $\mathbf{b}_i$ to form an affine matrix $\boldsymbol{\theta}$. We can see that for some loss function $L$ at layer $n$, the derivative with respect to parameters $\boldsymbol{\theta}_i$ is:

$$\frac{\partial L}{\partial \boldsymbol{\theta}_i} = \frac{\partial \mathbf{a}_{n+1}}{\partial \boldsymbol{\theta}_i} \frac{\partial L}{\partial \mathbf{a}_{n+1}} \tag{2}$$

The partial derivatives for the pre-activations can be decomposed as follows:

$$\begin{aligned} \frac{\partial \mathbf{a}_{i+1}}{\partial \boldsymbol{\theta}_i} &= \frac{\partial \mathbf{a}_i}{\partial \boldsymbol{\theta}_i} \frac{\partial \mathbf{h}_i}{\partial \mathbf{a}_i} \frac{\partial \mathbf{a}_{i+1}}{\partial \mathbf{h}_i} \\ &= \frac{\partial \mathbf{a}_i}{\partial \boldsymbol{\theta}_i} \, \mathbf{D}_i \mathbf{W}_{i+1} \; \rightarrow \; \frac{\partial \mathbf{a}_{i+1}}{\partial \mathbf{a}_i} = \mathbf{D}_i \mathbf{W}_{i+1}, \end{aligned} \tag{3}$$

where $\mathbf{D_i}$ is the Jacobian corresponding to the activation function, containing partial derivatives of the hidden units at layer $i + 1$ with respect to the pre-activation inputs. Typically, $\mathbf{D}$ is diagonal. Following the above, the gradient in equation 2 can be fully decomposed into a recursive chain of matrix products:

$$\frac{\partial L}{\partial \boldsymbol{\theta}_i} = \frac{\partial \mathbf{a}_i}{\partial \boldsymbol{\theta}_i} \prod_{j=i}^{n} (\mathbf{D}_j \mathbf{W}_{j+1}) \frac{\partial L}{\partial \mathbf{a}_{n+1}} \tag{4}$$

In (Pascanu et al., 2013), it is shown that the 2-norm of $\dfrac{\partial \mathbf{a}_{i+1}}{\partial \mathbf{a}_i}$ is bounded by the product of the norms of the non-linearity's Jacobian and transition matrix at time $t$ (layer $i$), as follows:

$$\left\| \frac{\partial \mathbf{a}_{t+1}}{\partial \mathbf{a}_t} \right\| \le \|\mathbf{D}_t\| \, \|\mathbf{W}_t\| \le \lambda_{\mathbf{D}_t} \, \lambda_{\mathbf{W}_t} = \eta_t,$$

$$\lambda_{\mathbf{D}_t}, \lambda_{\mathbf{W}_t} \in \mathbb{R}. \tag{5}$$

where $\lambda_{\mathbf{D}_t}$ and $\lambda_{\mathbf{W}_t}$ are the largest singular values of the non-linearity's Jacobian $\mathbf{D}_t$ and the transition matrix $\mathbf{W}_t$. In RNNs, $\mathbf{W}_t$ is shared across time and can be simply denoted as $\mathbf{W}$.

Equation 5 shows that the gradient can grow or shrink at each layer depending on the gain of each layer's linear transformation $\mathbf{W}$ and the gain of the Jacobian $\mathbf{D}$. The gain caused by each layer is magnified across all time steps or layers. It is easy to have extreme amplification in a recurrent neural network where $\mathbf{W}$ is shared across time steps and a non-unitary gain in $\mathbf{W}$ is amplified exponentially. The phenomena of extreme growth or contraction of the gradient across time steps or layers are known as the exploding and the vanishing gradient problems, respectively. It is sufficient for RNNs to have $\eta_t \le 1$ at each time $t$ to enable the possibility of vanishing gradients, typically for some large number of time steps $T$. The rate at which a gradient (or forward signal) vanishes

depends on both the parameterization of the model and on the input data. The parameterization may be conditioned by placing appropriate constraints on $\mathbf{W}$. It is worth keeping in mind that the Jacobian $\mathbf{D}$ is typically contractive, thus tending to be norm-reducing) and is also data-dependent, whereas $\mathbf{W}$ can vary from being contractive to norm-preserving, to expansive and applies the same gain on the forward signal as on the back-propagated gradient signal.

## 2 OUR APPROACH

Vanishing and exploding gradients can be controlled to a large extent by controlling the maximum and minimum *gain* of $\mathbf{W}$. The maximum gain of a matrix $\mathbf{W}$ is given by the spectral norm which is given by

$$||\mathbf{W}||_2 = \max \left[ \frac{||\mathbf{W}\mathbf{x}||}{||\mathbf{x}||} \right]. \tag{6}$$

By keeping our weight matrix $\mathbf{W}$ close to orthogonal, one can ensure that it is close to a norm-preserving transformation (where the spectral norm is equal to one, but the minimum gain is also one). One way to achieve this is via a simple soft constraint or regularization term of the form:

$$\lambda \sum_i ||\mathbf{W}_i^T \mathbf{W}_i - \mathbf{I}||^2. \tag{7}$$

However, it is possible to formulate a more direct parameterization or factorization for $\mathbf{W}$ which permits *hard bounds* on the amount of expansion and contraction induced by $\mathbf{W}$. This can be achieved by simply parameterizing $\mathbf{W}$ according to its singular value decomposition, which consists of the composition of orthogonal basis matrices $\mathbf{U}$ and $\mathbf{V}$ with a diagonal spectral matrix $\mathbf{S}$ containing the singular values which are real and positive by definition. We have

$$\mathbf{W} = \mathbf{U}\mathbf{S}\mathbf{V}^T. \tag{8}$$

Since the spectral norm or maximum gain of a matrix is equal to its largest singular value, this decomposition allows us to control the maximum gain or expansivity of the weight matrix by controlling the magnitude of the largest singular value. Similarly, the minimum gain or contractivity of a matrix can be obtained from the minimum singular value.

We can keep the bases $\mathbf{U}$ and $\mathbf{V}$ orthogonal via *geodesic gradient descent* along the set of weights that satisfy $\mathbf{U}^T\mathbf{U} = \mathbf{I}$ and $\mathbf{V}^T\mathbf{V} = \mathbf{I}$ respectively. The submanifolds that satisfy these constraints are called Stiefel manifolds. We discuss how this is achieved in more detail below, then discuss our construction for bounding the singular values.

During optimization, in order to maintain the orthogonality of an orthogonally-initialized matrix $\mathbf{M}$, i.e. where $\mathbf{M} = \mathbf{U}$, $\mathbf{M} = \mathbf{V}$ or $\mathbf{M} = \mathbf{W}$ if so desired, we employ a Cayley transformation of the update step onto the Stiefel manifold of (semi-)orthogonal matrices, as in Nishimori (2005) and Tagare (2011). Given an orthogonally-initialized parameter matrix $\mathbf{M}$ and its Jacobian, $\mathbf{G}$ with respect to the objective function, an update is performed as follows:

$$\begin{aligned} \mathbf{A} &= \mathbf{G}\mathbf{M}^T - \mathbf{M}\mathbf{G}^T \\ \mathbf{M}_{new} &= \mathbf{M} + (\mathbf{I} + \frac{\eta}{2}\mathbf{A})^{-1}(\mathbf{I} - \frac{\eta}{2}\mathbf{A}), \end{aligned} \tag{9}$$

where $\mathbf{A}$ is a skew-symmetric matrix (that depends on the Jacobian and on the parameter matrix) which is mapped to an orthogonal matrix via a Cayley transform and $\eta$ is the learning rate.

While the update rule in (9) allows us to maintain an orthogonal hidden to hidden transition matrix $\mathbf{W}$ if desired, we are interested in exploring the effect of stepping away from the Stiefel manifold. As such, we parameterize the transition matrix $\mathbf{W}$ in factorized form, as a singular value decomposition with orthogonal bases $\mathbf{U}$ and $\mathbf{V}$ updated by geodesic gradient descent using the Cayley transform approach above.

If $\mathbf{W}$ is an orthogonal matrix, the singular values in the diagonal matrix $\mathbf{S}$ are all equal to one. However, in our formulation we allow these singular values to deviate from one and employ a sigmoidal parameterization to apply a hard constraint on the maximum and minimum amount of

deviation. Specifically, we define a margin $m$ around 1 within which the singular values must lie. This is achieved with the parameterization

$$s_i = 2m(\sigma(p_i) - 0.5) + 1, \qquad s_i \in \{\text{diag}(\mathbf{S})\}, \; m \in [0, \; 1]. \tag{10}$$

The singular values are thus restricted to the range $[1 - m, \; 1 + m]$ and the underlying parameters $p_i$ are updated freely via stochastic gradient descent. Note that this parameterization strategy also has implications on the step sizes that gradient descent based optimization will take when updating the singular values – they tend to be smaller compared to models with no margin constraining their values. Specifically, a singular value's progression toward a margin is slowed the closer it is to the margin. The sigmoidal parameterization can also impart another effect on the step size along the spectrum which needs to be accounted for. Considering 10, the gradient backpropagation of some loss $L$ toward parameters $p_i$ is found as

$$\frac{dL}{dp_i} = \frac{ds_i}{dp_i}\frac{dL}{ds_i} = 2m\frac{d\sigma(p_i)}{dp_i}\frac{dL}{ds_i}. \tag{11}$$

From (11), it can be seen that the magnitude of the update step for $p_i$ is scaled by the margin hyperparameter $m$. This means for example that for margins less than one, the effective learning rate for the spectrum is reduced in proportion to the margin. Consequently, we adjust the learning rate along the spectrum to be independent of the margin by renormalizing it by $2m$.

This margin formulation both guarantees singular values lie within a well defined range and slows deviation from orthogonality. Alternatively, one could enforce the orthogonality of $\mathbf{U}$ and $\mathbf{V}$ and impose a regularization term corresponding to a mean one Gaussian prior on these singular values. This encourages the weight matrix $\mathbf{W}$ to be norm preserving with a controllable strength equivalent to the variance of the Gaussian. We also explore this approach further below.

## 3 EXPERIMENTS

In this section, we explore hard and soft orthogonality constraints on factorized weight matrices for recurrent neural network hidden to hidden transitions. With hard orthogonality constraints on $\mathbf{U}$ and $\mathbf{V}$, we investigate the effect of widening the spectral margin or bounds on convergence and performance. Loosening these bounds allows increasingly larger margins within which the transition matrix $\mathbf{W}$ can deviate from orthogonality. We confirm that orthogonal initialization is useful as noted in Henaff et al. (2016), and we show that although strict orthogonality guarantees stable gradient norm, loosening orthogonality constraints can increase the rate of gradient descent convergence. We begin our analyses on tasks that are designed to stress memory: a sequence copying task and a basic addition task (Hochreiter & Schmidhuber, 1997). We then move on to tasks on real data that require models to capture long-range dependencies: digit classification based on sequential and permuted MNIST vectors (Le et al., 2015; LeCun et al., 1998). Finally, we look at a basic language modeling task using the Penn Treebank dataset (Marcus et al., 1993).

The copy and adding tasks, introduced by Hochreiter & Schmidhuber (1997), are synthetic benchmarks with pathologically hard long distance dependencies that require long-term memory in models. The copy task consists of an input sequence that must be remembered by the network, followed by a series of blank inputs terminated by a delimiter that denotes the point at which the network must begin to output a copy of the initial sequence. We use an input sequence of $T + 20$ elements that begins with a sub-sequence of 10 elements to copy, each containing a symbol $a_i \in \{a_1, ..., a_p\}$ out of $p = 8$ possible symbols. This sub-sequence is followed by $T - 1$ elements of the blank category $a_0$ which is terminated at step $T$ by a delimiter symbol $a_{p+1}$ and 10 more elements of the blank category. The network must learn to remember the initial 10 element sequence for $T$ time steps and output it after receiving the delimiter symbol.

The goal of the adding task is to add two numbers together after a long delay. Each number is randomly picked at a unique position in a sequence of length $T$. The sequence is composed of $T$ values sampled from a uniform distribution in the range $[0, 1)$, with each value paired with an indicator value that identifies the value as one of the two numbers to remember (marked 1) or as a value to ignore (marked 0). The two numbers are positioned randomly in the sequence, the first in the range $[0, \frac{T}{2} - 1]$ and the second in the range $[\frac{T}{2}, T - 1]$, where 0 marks the first element. The network must learn to identify and remember the two numbers and output their sum.

The sequential MNIST task from Le et al. (2015), MNIST digits are flattened into vectors that can be traversed sequentially by a recurrent neural network. The goal is to classify the digit based on the sequential input of pixels. The simple variant of this task is with a simple flattening of the image matrices; the harder variant of this task includes a random permutation of the pixels in the input vector that is determined once for an experiment. The latter formulation introduces longer distance dependencies between pixels that must be interpreted by the classification model.

The English Penn Treebank (PTB) dataset from Marcus et al. (1993) is an annotated corpus of English sentences, commonly used for benchmarking language models. We employ a sequential character prediction task: given a sentence, a recurrent neural network must predict the next character at each step, from left to right. We use input sequences of variable length, with each sequence containing one sentence. We model 49 characters including lowercase letters (all strings are in lowercase), numbers, common punctuation, and an unknown character placeholder. In our experiments on two subsets of the data: in the first, we first use 23% of the data with strings with up to 75 characters and in the second we include over 99% of the dataset, picking strings with up to 300 characters.

## 3.1 LOOSENING HARD ORTHOGONALITY CONSTRAINTS

In this section, we experimentally explore the effect of loosening hard orthogonality constraints through loosening the spectral margin defined above for the hidden to hidden transition matrix.

In all experiments, we employed RMSprop (Tieleman & Hinton, 2012) when not using geodesic gradient descent. We used minibatches of size 50 and for generated data (the copy and adding tasks), we assumed an epoch length of 100 minibatches. We cautiously introduced gradient clipping at magnitude 100 (unless stated otherwise) in all of our RNN experiments although it may not be required and we consistently applied a small weight decay of 0.0001. Unless otherwise specified, we trained all simple recurrent neural networks with the hidden to hidden matrix factorization as in (8) using geodesic gradient descent on the bases (learning rate $10^{-6}$) and RMSprop on the other parameters (learning rate 0.0001), using a tanh transition nonlinearity, and clipping gradients of 100 magnitude. The neural network code was built on the Theano framework (Theano Development Team, 2016). When parameterizing a matrix in factorized form, we apply the weight decay on the composite matrix rather than on the factors in order to be consistent across experiments. For MNIST and PTB, test set metrics were computed based on the parameterization that gave the best validation set accuracy.

### 3.1.1 CONVERGENCE ON SYNTHETIC MEMORY TASKS

For different sequence lengths $T$ of the copy and adding tasks, we trained a factorized RNN with 128 hidden units and various spectral margins $m$. For the copy task, we used Elman networks without a transition non-linearity as in Henaff et al. (2016). We discuss our investigations into the use of a non-linearity on the copy task in the Appendix.

As shown in Figure 1 we see an increase in the rate of convergence as we increase the spectral margin. This observation generally holds across the tested sequence lengths ($T = 200$, $T = 500$, $T = 1000$, $T = 10000$); however, large spectral margins hinder convergence on extremely long sequence lengths. At sequence length $T = 10000$, parameterizations with spectral margins larger than 0.001 converge slower than when using a margin of 0.001. In addition, the experiment without a margin failed to converge on the longest sequence length. This follows the expected pattern where stepping away from the Stiefel manifold may help with gradient descent optimization but loosening orthogonality constraints can reduce the stability of signal propagation through the network.

For the adding task, we trained a factorized RNN on $T = 1000$ length sequences, using a ReLU activation function on the hidden to hidden transition matrix. The mean squared error (MSE) is shown for different spectral margins in Figure 5 in the Appendix. Testing spectral margins $m = 0$, $m = 1$, $m = 10$, $m = 100$, and no margin, we find that the models with the purely orthogonal ($m = 0$) and the unconstrained (no margin) transition matrices failed to begin converging beyond baseline MSE within 2000 epochs.

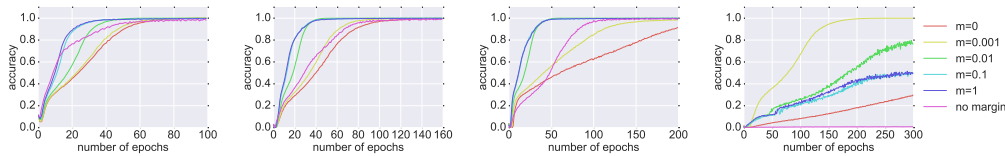

Figure 1: Accuracy curves on the copy task for sequence lengths of (from left to right) T=200, T=500, T=1000, T=10000 given different spectral margins. Convergence speed increases with margin size; however, large margin sizes are ineffective at longer sequence lengths (T=10000, right).

| margin | initialization | accuracy |
|---|---|---|
| 0 | orthogonal | 77.18 |
| 0.001 | orthogonal | 79.26 |
| 0.01 | orthogonal | 85.47 |
| 0.1 | orthogonal | 94.10 |
| 1 | orthogonal | 93.84 |
| none | orthogonal | 93.24 |
| none | Glorot normal | 66.71 |
| none | identity | 53.53 |
| | LSTM | 97.30 |

Table 1: Ordered sequential MNIST classification with different margin sizes and an LSTM.

| margin | initialization | accuracy |
|---|---|---|
| 0 | orthogonal | 83.56 |
| 0.001 | orthogonal | 84.59 |
| 0.01 | orthogonal | 89.63 |
| 0.1 | orthogonal | 91.44 |
| 1 | orthogonal | 90.83 |
| none | orthogonal | 90.51 |
| none | Glorot normal | 79.33 |
| none | identity | 42.72 |
| | LSTM | 92.62 |

Table 2: Permuted sequential MNIST classification with different margin sizes and an LSTM.

### 3.1.2 PERFORMANCE ON REAL DATA

Having confirmed that an orthogonality constraint can negatively impact convergence rate, we seek to investigate the effect on model performance for tasks on real data. We show the results of experiments on permuted sequential MNIST in Table 2 and ordered sequential MNIST in Table 1. The loss curves are shown in Figure 6 in the Appendix and reveal an increased convergence rate for larger spectral margins. We trained the factorized RNN models with 128 hidden units for 120 epochs. We also trained an LSTM with 128 hidden units on both tasks for 150 epochs, configured with peephole connections, orthogonally initialized (and forget gate bias initialized to one), and trained with RMSprop (learning rate 0.0001, clipping gradients of magnitude 1).

We show the results of experiments on PTB character prediction, in terms of bits per character (bpc) and prediction accuracy, for a subset of short sequences (up to 75 characters; 23% of data) in Table 3 and for a subset of long sequences (up to 300 characters; 99% of data) in Table 4. We trained factorized RNN models with 512 hidden units for 200 epochs with geodesic gradient descent on the bases (learning rate $10^{-6}$) and RMSprop on the other parameters (learning rate 0.001), using a tanh transition nonlinearity, and clipping gradients of 30 magnitude.

Interestingly, for both the ordered and permuted sequential MNIST tasks, models with a non-zero margin significantly outperform those that are constrained to have purely orthogonal transition matri-

| margin | initialization | bpc | accuracy |
|---|---|---|---|
| 0 | orthogonal | 2.16 | 55.31 |
| 0.01 | orthogonal | 2.16 | 55.33 |
| 0.1 | orthogonal | 2.12 | 55.37 |
| 1 | orthogonal | 2.06 | 57.07 |
| 100 | orthogonal | 2.04 | 57.51 |
| none | orthogonal | 2.06 | 57.38 |
| none | Glorot normal | 2.08 | 57.37 |
| none | identity | 2.25 | 53.83 |

Table 3: Character prediction on PTB sentences of to 75 characters, using different margins.

| margin | initialization | bpc | accuracy |
|---|---|---|---|
| 0 | orthogonal | 2.20 | 54.88 |
| 0.01 | orthogonal | 2.20 | 54.83 |
| 0.1 | orthogonal | 2.24 | 54.10 |
| 1 | orthogonal | 2.36 | 51.12 |
| 100 | orthogonal | 2.36 | 51.20 |
| none | orthogonal | 2.34 | 51.30 |
| none | Glorot normal | 2.34 | 51.04 |
| none | identity | 2.68 | 45.35 |

Table 4: Character prediction on PTB sentences of up to 300 characters, using different margins.

ces (margin of zero). The best results on both the ordered and sequential MNIST tasks were yielded by models with a spectral margin of 0.1, at 94.10% accuracy and 91.44% accuracy, respectively. An LSTM outperformed the RNNs in both tasks; nevertheless, RNNs with hidden to hidden transitions initialized as orthogonal matrices performed admirably without a memory component and without all of the additional parameters associated with gates. Indeed, orthogonally initialized RNNs performed almost on par with the LSTM in the permuted sequential MNIST task which presents longer distance dependencies than the ordered task. Although the optimal margin appears to be 0.1, RNNs with large margins perform almost identically to an RNN without a margin, as long as the transition matrix is initialized as orthogonal. On these tasks, orthogonal initialization appears to significantly outperform Glorot normal initialization (Glorot & Bengio, 2010) or initializing the matrix as identity. It is interesting to note that for the MNIST tasks, orthogonal initialization appears useful while orthogonality constraints appear mainly detrimental. This suggests that while orthogonality helps early training by stabilizing gradient flow across many time steps, orthogonality constraints may need to be loosened on some tasks so as not to over-constrain the model's representational ability.

Curiously, larger margins and even models without sigmoidal constraints on the spectrum (no margin) performed well as long as they were initialized to be orthogonal, suggesting that evolution away from orthogonality is not a serious problem on MNIST. It is not surprising that orthogonality is useful for the MNIST tasks since they depend on long distance signal propagation with a single output at the end of the input sequence. On the other hand, character prediction with PTB produces an output at every time step. Constraining deviation from orthogonality proved detrimental for short sentences (Table 3) and beneficial when long sentences were included (Table 4). Furthermore, Glorot normal initialization did not perform worse than orthogonal initialization for PTB. Since an output is generated for every character in a sentence, short distance signal propagation is possible. Thus it is possible that the RNN is first learning very local dependencies between neighbouring characters and that given enough context, constraining deviation from orthogonality can help force the network to learn longer distance dependencies.

### 3.1.3 SPECTRAL AND GRADIENT EVOLUTION

It is interesting to note that even long sequence lengths (T=1000) in the copy task can be solved efficiently with rather large margins on the spectrum. In Figure 2 we look at the gradient propagation of the loss from the last time step in the network with respect to the hidden activations. We can see that for a purely orthogonal parameterization of the transition matrix (when the margin is zero), the gradient norm is preserved across time steps, as expected. We further observe that with increasing margin size, the number of update steps over which this norm preservation survives decreases, though surprisingly not as quickly as expected.

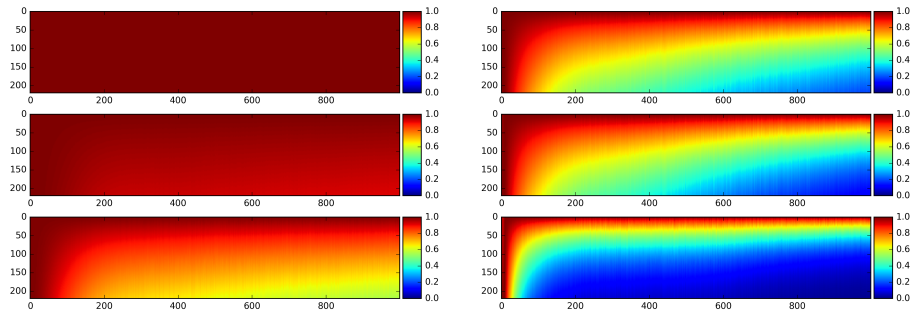

Figure 2: The norm of the gradient of the loss from the last time step with respect to the hidden units at a given time step for a length 220 RNN over 1000 update iterations for different margins. Iterations are along the abscissa and time steps are denoted along the ordinate. The first column margins are: 0, 0.001, 0.01. The second column margins are: 0.1, 1, no margin. Gradient norms are normalized across the time dimension.

Although the deviation of singular values from one should be slowed by the sigmoidal parameterizations, even parameterizations without a sigmoid (no margin) can be effectively trained for all but the longest sequence lengths. This suggests that the spectrum is not deviating far from orthogonality and that inputs to the hidden to hidden transitions are mostly not aligned along the dimensions of great-

est expansion or contraction. We evaluated the spread of the spectrum in all of our experiments and found that indeed, singular values tend to stay well within their prescribed bounds and only reach the margin when using a very large learning rate that does not permit convergence. Furthermore, when transition matrices are initialized as orthogonal, singular values remain near one throughout training even without a sigmoidal margin for tasks that require long term memory (copy, adding, sequential MNIST). On the other hand, singular value distributions tend to drift away from one for PTB character prediction which may help explain why enforcing an orthogonality constraint can be helpful for this task, when modeling long sequences. Interestingly, singular values spread out less for longer sequence lengths (nevertheless, the T=10000 copy task could not be solved with no sigmoid on the spectrum).

We visualize the spread of singular values for different model parameterizations on the permuted sequential MNIST task in Figure 3. Curiously, we find that the distribution of singular values tends to shift upward to a mean of approximately 1.05 on both the ordered and permuted sequential MNIST tasks. We note that in those experiments, a tanh transition nonlinearity was used which is contractive in both the forward signal pass and the gradient backward pass. An upward shift in the distribution of singular values of the transition matrix would help compensate for that contraction. Indeed, (Saxe et al., 2013) describe this as a possibly good regime for learning in deep neural networks. That the model appears to evolve toward this regime suggests that deviating from it may incur a cost. This is interesting because the cost function cannot take into account numerical issues such as vanishing or exploding gradients (or forward signals); we do not know what could make this deviation costly. That the transition matrix may be compensating for the contraction of the tanh is supported by further experiments: applying a 1.05 pre-activation gain appears to allow a model with a margin of 0 to nearly match the top performance reached on both of the MNIST tasks. Furthermore, when using the OPLU norm-preserving activation function (Chernodub & Nowicki, 2016), we found that orthogonally initialized models performed equally well with all margins, achieving over 90% accuracy on the permuted sequential MNIST task. Unlike orthgonally initialized models, the RNN on the bottom right of Figure 3 with Glorot normal initialized transition matrices, begins and ends with a wide singular spectrum. While there is no clear positive shift in the distribution of singular values, the mean value appears to very gradually increase for both the ordered and permuted sequential MNIST tasks. If the model is to be expected to positively shift singular values to compensate for the contractivity of the tanh nonlinearity, it is not doing so well for the Glorot-initialized case; however, this may be due to the inefficiency of training as a result of vanishing gradients, given that initialization.

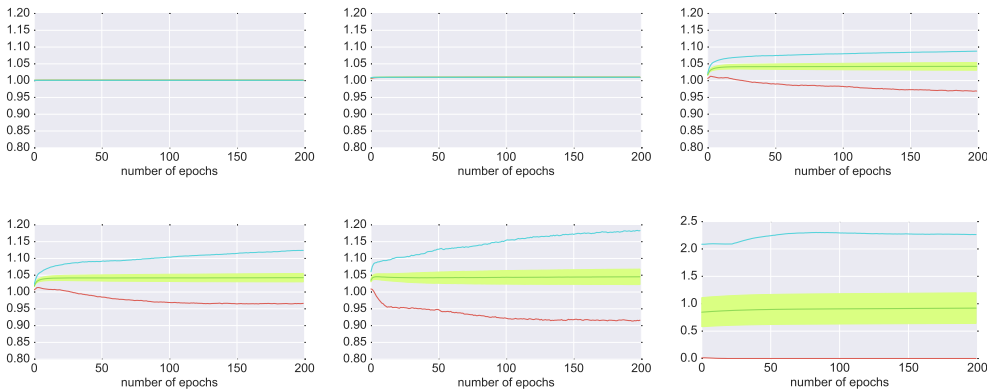

Figure 3: Singular value evolution on the permuted sequential MNIST task for factorized RNNs with different margin sizes. Margins are, from left to right: *top row*: 0.001, 0.01, 0.1; *bottom row*: 1, no margin, no margin. The singular value distributions are summarized with the mean (green line, center) and standard deviation (green shading about mean), minimum (red, bottom) and maximum (blue, top) values. All models are initialized with orthogonal hidden to hidden transition matrices except for the model on the bottom right where Glorot normal initialization is used.

## 3.2 Exploring soft orthogonality constraints

Having established that it may indeed be useful to step away from orthogonality, here we explore two forms of soft constraints (rather than hard bounds as above) on hidden to hidden transition matrix orthogonality. The first is a simple penalty that directly encourages a transition matrix $\mathbf{W}$ to be orthogonal, of the form $\lambda||\mathbf{W}^T\mathbf{W} - \mathbf{I}||_2^2$. This is similar to the orthogonality penalty introduced by Henaff et al. (2016). In the first two subfigures on the left of Figure 4, we explore the effect of weakening this form of regularization. We trained both a regular non-factorized RNN on the $T = 200$ copy task and a factorized RNN with orthogonal bases on the $T = 500$ copy task. For the regular RNN, we had to reduce the learning rate to $10^{-5}$. Here again we see that weakening the strength of the orthogonality-encouraging penalty can increase convergence speed.

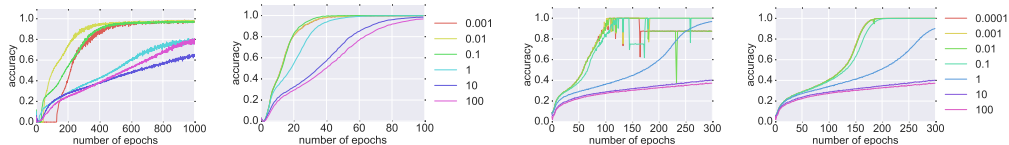

Figure 4: Accuracy curves on the copy task for different strengths of soft orthogonality constraints. A soft orthogonality constraint is applied to the transition matrix $\mathbf{W}$ for a regular RNN on $T = 200$ (Left) and the same is applied on a factorized RNN on $T = 500$ (Left center). Another constraint in the form of a mean one Gaussian prior on the singular values is applied to a factorized RNN on $T = 200$ (Right center); the same is applied to a factorized RNN with a sigmoidal parameterization of the spectrum, using a large margin of 1 (Right). Loosening orthogonality speeds convergence.

The second approach we explore replaces the sigmoidal margin parameterization with a mean one Gaussian prior on the singular values. In the two right subfigures of Figure 4, we visualize the accuracy on the length 200 copy task, using geoSGD (learning rate $10^{-6}$) to keep $\mathbf{U}$ and $\mathbf{V}$ orthogonal and different strengths of a Gaussian prior with mean one on the singular values. We trained these experiments with regular SGD on the spectrum and other non-orthogonal parameter matrices, using a $10^{-5}$ learning rate. We see that priors which are too strong lead to slow convergence. Loosening the strength of the prior makes the optimization more efficient. Furthermore, we compare a direct parameterization of the spectrum (no sigmoid) in Figure 4 with a sigmoidal parameterization, using a large margin of 1. Without the sigmoidal parameterization, optimization quickly becomes unstable; on the other hand, the optimization also becomes unstable if the prior is removed completely in the sigmoidal formulation (margin 1). These results further motivate the idea that parameterizations that deviate from orthogonality may perform better than purely orthogonal ones, as long as they are sufficiently constrained to avoid instability during training.

## 4 Conclusions

We have explored a number of methods for controlling the expansivity of gradients during backpropagation based learning in RNNs through manipulating orthogonality constraints and regularization on matrices. Our experiments indicate that while orthogonal initialization may be beneficial, maintaining constraints on orthogonality can be detrimental. Indeed, moving away from hard constraints on matrix orthogonality can help improve optimization convergence rate and model performance. However, we also observe with synthetic tasks that relaxing regularization which encourages the spectral norms of weight matrices to be close to one, or allowing bounds on the spectral norms of weight matrices to be too wide, can reverse these gains and may lead to unstable optimization.

### Acknowledgments

We thank the Natural Sciences and Engineeering Research Council (NSERC) of Canada and Samsung for supporting this research.

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

## 5 APPENDIX

### 5.1 ADDITIONAL FIGURES

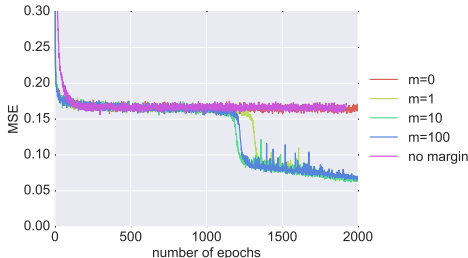

Figure 5: Mean squared error (MSE) curves on the adding task for different spectral margins $m$. For a trivial baseline solution of always outputting the same number, the expected baseline MSE is 0.167.

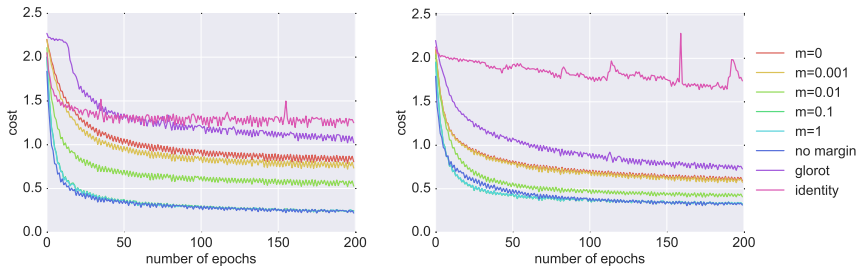

Figure 6: Loss curves for different factorized RNN parameterizations on the sequential MNIST task (left) and the permuted sequential MNIST task (right). The spectral margin is denoted by m; models with no margin have singular values that are directly optimized with no constraints; Glorot refers to a factorized RNN with no margin that is initialized with Glorot normal initialization.

### 5.2 COPY TASK NONLINEARITY

We found that nonlinearities such as a rectified linear unit (ReLU) (Nair & Hinton, 2010) or hyperbolic tangent (tanh) made the copy task far more difficult to solve. Using tanh, a short sequence length ($T = 100$) copy task required both a soft constraint that encourages orthogonality and thousands of epochs for training. It is worth noting that in the unitary evolution recurrent neural network of Arjovsky et al. (2015), the non-linearity (referred to as the "modReLU") is actually initialized as an identity operation that is free to deviate from identity during training. Furthermore, Henaff et al. (2016) derive a solution mechanism for the copy task that drops the non-linearity from an RNN. To explore this further, we experimented with a parametric leaky ReLU activation function (PReLU) which introduces a trainable slope $\alpha$ for negative valued inputs $x$, producing $f(x) = max(x, 0) + \alpha min(x, 0)$ (He et al., 2015). Setting the slope $\alpha$ to one would make the PReLU equivalent to an identity function. We experimented with clamping $\alpha$ to 0.5, 0.7 or 1 in a factorized RNN with a spectral margin of 0.3 and found that only the model with $\alpha = 1$ solved the $T = 1000$ length copy task. We also experimented with a trainable slope $\alpha$, initialized to 0.7 and found that it converges to 0.96, further suggesting the optimal solution for the copy task is without a transition nonlinearity. Since the copy task is purely a memory task, one may imagine that a transition nonlinearity such as a tanh or ReLU may be detrimental to the task as it can lose information. Thus, we also tried a recent activation function that preserves information, called an orthogonal permutation linear unit (OPLU) (Chernodub & Nowicki, 2016). The OPLU preserves norm, making a fully norm-preserving RNN possible. Interestingly, this activation function allowed us to recover identical results on the copy task to those without a nonlinearity for different spectral margins.

## 5.3 METHOD RUNNING TIME

Although the method proposed in section 2 relies on a matrix inversion, an operation with $O(n^3)$ complexity for an n × n matrix, the running time of an RNN factorized in such a way actually remains reasonable. This running time is summarized in Table 5 and includes all computations in the graph, together with the matrix inversion. As this method is meant to be used only for the analysis in this work, we find the running times acceptable for that purpose. Models were run on an Nvidia GTX-770 GPU and were run against the T=100 length copy task.

| hidden units | SGD | geoSGD |
|---|---|---|
| 128 | $21.9 \pm 0.2$ | $40.4 \pm 0.1$ |
| 500 | $46.7 \pm 0.2$ | $161.4 \pm 0.2$ |
| 1000 | $95.4 \pm 0.3$ | $711.2 \pm 0.8$ |

Table 5: Run time in seconds for 1000 iterations on a T=100 copy task of a regular RNN trained with stochastic gradient descent (SGD) compared against a factorized RNN trained with geodesic SGD on the bases (geoSGD) and regular SGD for other parameters.

