# Peer review of "On orthogonality and learning recurrent networks with long term dependencies"

_ICLR 2017 — rejected_

[Official Review · AnonReviewer2 · rating 5 · confidence 5 · 15 Dec 2016]
**This paper investigates the issue of orthogonality of the transfer weight matrix in RNNs and suggests an optimization formulation on the manifold of (semi)orthogonal matrices.**

Vanishing and exploding gradients makes the optimization of RNNs very challenging. The issue becomes worse on tasks with long term dependencies that requires longer RNNs. One of the suggested approaches to improve the optimization is to optimize in a way that the transfer matrix is almost orthogonal. This paper investigate the role of orthogonality on the optimization and learning which is very important. The writing is sound and clear and arguments are easy to follow. The suggested optimization method is very interesting. The main shortcoming of this paper is the experiments which I find very important and I hope authors can update the experiment section significantly. Below I mention some comments on the experiment section:

1- I think the experiments are not enough. At the very least, report the result on the adding problem and language modeling task on Penn Treebank.

2- I understand that the copying task becomes difficult with non-lineary. However, removing non-linearity makes the optimization very different and therefore, it is very hard to conclude anything from the results on the copying task.

3- I was not able to find the number of hidden units used for RNNs in different tasks.

4- Please report the running time of your method in the paper for different numbers of hidden units, compare it with the SGD and mention the NN package you have used.

5- The results on Table 1 and Table 2 might also suggest that the orthogonality is not really helpful since even without a margin, the numbers are very close compare to the case when you find the optimal margin. Am I right?

6- What do we learn from Figure 2? It is left without any discussion.

[Official Review · AnonReviewer3 · rating 5 · confidence 4 · 19 Dec 2016]
**Interesting question and proposed approach, with significance restricted by limited experimental settings.**

The paper is well-motivated, and is part of a line of recent work investigating the use of orthogonal weight matrices within recurrent neural networks. While using orthogonal weights addresses the issue of vanishing/exploding gradients, it is unclear whether anything is lost, either in representational power or in trainability, by enforcing orthogonality. As such, an empirical investigation that examines how these properties are affected by deviation from orthogonality is a useful contribution.

The paper is clearly written, and the primary formulation for investigating soft orthogonality constraints (representing the weight matrices in their SVD factorized form, which gives explicit control over the singular values) is clean and natural, albeit not necessarily ideal from a practical computational standpoint (as it requires maintaining multiple orthogonal weight matrices each requiring an expensive update step). I am unaware of this approach being investigated previously.

The experimental side, however, is somewhat lacking. The paper evaluates two tasks: a copy task, using an RNN architecture without transition non-linearities, and sequential/permuted sequential MNIST. These are reasonable choices for an initial evaluation, but are both toy problems and don't shed much light on the practical aspects of the proposed approaches. An evaluation in a more realistic setting would be valuable (e.g., a language modeling task).

Furthermore, while investigating pure RNN's makes sense for evaluating effects of orthogonality, it feels somewhat academic: LSTMs also provide a mechanism to capture longer-term dependencies, and in the tasks where the proposed approach was compared directly to an LSTM, it was significantly outperformed. It would be very interesting to see the effects of the proposed soft orthogonality constraint in additional architectures (e.g., deep feed-forward architectures, or whether there's any benefit when embedded within an LSTM, although this seems doubtful).

Overall, the paper addresses a clear-cut question with a well-motivated approach, and has interesting findings on some toy datasets. As such I think it could provide a valuable contribution. However, the significance of the work is restricted by the limited experimental settings (both datasets and network architectures).

[Official Review · AnonReviewer1 · rating 7 · confidence 4 · 20 Dec 2016]
**Interesting investigation into orthogonal parametrizations and initializations for RNNs**

This paper investigates the impact of orthogonal weight matrices on learning dynamics in RNNs. The paper proposes a variety of interesting optimization formulations that enforce orthogonality in the recurrent weight matrix to varying degrees. The experimental results demonstrate several conclusions: enforcing exact orthogonality does not help learning, while enforcing soft orthogonality or initializing to orthogonal weights can substantially improve learning. While some of the optimization methods proposed currently require matrix inversion and are therefore slow in wall clock time, orthogonal initialization and some of the soft orthogonality constraints are relatively inexpensive and may find their way into practical use.

The experiments are generally done to a high standard and yield a variety of useful insights, and the writing is clear.

The experimental results are based on using a fixed learning rate for the different regularization strengths. Learning speed might be highly dependent on this, and different strengths may admit different maximal stable learning rates. It would be instructive to optimize the learning rate for each margin separately (maybe on one of the shorter sequence lengths) to see how soft orthogonality impacts the stability of the learning process. Fig. 5, for instance, shows that a sigmoid improves stability—but perhaps slightly reducing the learning rate for the non-sigmoid Gaussian prior RNN would make the learning well-behaved again for weightings less than 1.

Fig. 4 shows singular values converging around 1.05 rather than 1. Does initializing to orthogonal matrices multiplied by 1.05 confer any noticeable advantage over standard orthogonal matrices? Especially on the T=10K copy task?

“Curiously, larger margins and even models without sigmoidal constraints on the spectrum (no margin) performed well as long as they were initialized to be orthogonal suggesting that evolution away from orthogonality is not a serious problem on this task.” This is consistent with the analysis given in Saxe et al. 2013, where for deep linear nets, if a singular value is initialized to 1 but dies away during training, this is because it must be zero to implement the desired input-output map. More broadly, an open question has been whether orthogonality is useful as an initialization, as proposed by Saxe et al., where its role is mainly as a preconditioner which makes optimization proceed quickly but doesn’t fundamentally change the optimization problem; or whether it is useful as a regularizer, as proposed by Arjovsky et al. 2015 and Henaff et al. 2015, that is, as an additional constraint in the optimization problem (minimize loss subject to weights being orthogonal). These experiments seem to show that mere initialization to orthogonal weights is enough to reap an optimization speed advantage, and that too much regularization begins to hurt performance—i.e., substantially changing the optimization problem is undesirable. This point is also apparent in Fig. 2: In terms of the training loss on MNIST (Fig. 2), no margin does almost indistinguishably from a margin of 1 or .1. However in terms of accuracy, a margin of .1 is best. This shows that large or nonexistent margins (i.e., orthogonal initializations) enable fast optimization of the training loss, but among models that attain similar training loss, the more nearly orthogonal weights perform better. This starts to separate out the optimization speed advantage conferred by orthogonality from the regularization advantage it confers. It may be useful to more explicitly discuss the initialization vs regularization dimension in the text.

Overall, this paper contributes a variety of techniques and intuitions which are likely to be useful in training RNNs.

[Author Response · Eugene Vorontsov · 19 Jan 2017]
**Summary of additional experiments**

As requested, we have performed a number of additional experiments and added them to the paper draft. The new experiments can be summarized as follows:

[Adding task]
We explored the use of different spectral margins on a synthetic adding task with a transition nonlinearity. A purely orthogonal model (margin=0) and one with no margin failed to converge beyond the baseline level on this task; however, models with non-zero margins began to converge beyond the baseline.

[Penn Treebank]
We have explored the use of different spectral margins for character prediction on Penn Treebank using two setups. (1) A subset (about 23%) of the data containing sentences with up to 75 characters. (2) A subset (over 99%) of the data containing sentences with up to 300 characters.

[Parametric ReLU on the copy task]
We explored the use of a trainable non-linearity on the copy task and found that it learns to nearly become an identity function. We also experimented with manually setting a leaky ReLU. (This is in the Appendix)

[OPLU]
We have tested a norm-preserving activation function on the copy task and on the MNIST tasks. We found that it allows an RNN’s performance on the copy task to match that of an RNN without a nonlinearity. On MNIST, it allows all spectral margins to perform equally well. We discuss this further in the draft.

[MNIST with an orthogonal RNN + gain]
As suggested following the observation that the singular spectra tend toward distributions with mean 1.05 on the MNIST tasks, we trained an RNN with a transition matrix constrained to be purely orthogonal and applied a 1.05 gain on the hidden pre-activations. We found that this allowed purely orthogonal models to nearly match the top scores on these tasks instead of underperforming as they do without that gain.

[Final Decision · Program Chairs · 06 Feb 2017]
**ICLR committee final decision**

The work explores a very interesting way to deal with the problem of propagating information through RNNs. I think the approach looks promising but the reviewers point out that the experimental evaluation is a bit lacking. In particular, it focuses on simple problems and does not demonstrate a clear advantage over LSTMs. I would encourage the authors to continue pursuing this direction, but without a theoretical advance I believe that additional empirical evidence would still be needed here.